# Screening and Genomic Analysis of *Bacillus velezensis* R12 as a Biocontrol Agent Against *Fusarium oxysporum* Causing Wilt in Longya Lily (*Lilium brownii* var. *viridulum*)

**DOI:** 10.3390/microorganisms13112430

**Published:** 2025-10-23

**Authors:** Huiying Guo, Yushan Huang, Zihan Cheng, Qinyuan Zheng, Qingxiu Li, Mengting Zhan, Hongjie Ji, Kuan Zhao, Du Zhu, Shaofang Liu

**Affiliations:** 1Key Laboratory of Natural Microbial Medicine Research of Jiangxi Province, Jiangxi Science and Technology Normal University, Nanchang 330013, China; ghymx8023@163.com (H.G.); 15874903122@163.com (Y.H.); chengzihan0515@163.com (Z.C.); 18268730345@163.com (Q.Z.); 15145153629@163.com (Q.L.); lince_er@163.com (M.Z.); key1989@126.com (K.Z.); 2Key Laboratory of Microbial Resources and Metabolism of Nanchang City, Jiangxi Science and Technology Normal University, Nanchang 330013, China; 3National Key Laboratory of Uranium Resources Exploration-Mining and Nuclear Remote Sensing, East China University of Technology, Nanchang 330013, China; a13875759760@126.com

**Keywords:** *Bacillus velezensis*, Lily bulb rot, *Lilium brownii* var. *viridulum*, antagonistic bacteria, biological control

## Abstract

Longya Lily (*Lilium brownii* var. *viridulum*) bulb rot, a devastating soil-borne disease caused by *Fusarium oxysporum* f. sp. *lilii* (Fol L1-1), severely compromises yield and quality of this economically significant crop. In this study, strain R12 was isolated from the rhizosphere soil of asymptomatic Longya lily plants and identified as *Bacillus velezensis*. The strain significantly disrupted the spore germination and hyphal morphology of Fol L1-1. In pot experiments, R12 not only effectively suppressed disease development but also promoted plant growth, a trait potentially linked to its indole-3-acetic acid (IAA) production capacity. Genomic analysis revealed a 4,015,523 bp circular chromosome (46.42% GC content) harboring gene clusters for the synthesis of diverse secondary metabolites, including surfactin, fengycin, difficidin, and bacillibactin. These findings highlight the potential of *B. velezensis* R12 as a biocontrol agent and provide insights into its mechanisms for suppressing phytopathogens and promoting plant growth.

## 1. Introduction

Longya lily (*Lilium brownii* var. *viridulum*), a commercially significant geophyte endemic to subtropical China, is intensively cultivated in Hunan and Jiangxi provinces for its economic and medicinal value. Its bulbs exhibit a distinctive morphological profile, characterized by large size, waxy epidermis, and viscous parenchyma tissue when cooked, enabling dual applications in culinary traditions and phytotherapy. Phytochemical analyses confirm the presence of bioactive steroidal saponins, polysaccharides, alkaloids, phenolics, and volatile oils [1,2,3] which synergistically confer demonstrated pharmacological activities, including: antioxidant, anti-inflammatory, immunomodulatory, anti-tumour and hypoglycaemic activities [3]. Owing to its high nutrient density and remarkable therapeutic potential, it is designated “Southern Ginseng” in traditional herbalism [1].

However, the rapid intensification of Longya lily monoculture faces mounting constraints from soil-borne diseases, particularly a destructive bulb rot and systemic wilt caused by *Fusarium oxysporum* f. sp. *lilii* (Fol-L1-1). This pathogen, capable of long-term soil survival and vascular colonization, leads to substantial yield losses through impaired bulb maturation and post-harvest deterioration [4,5,6,7,8]. Compounding this threat, the limited availability of registered fungicides and the accelerating development of pathogen resistance underscore the urgent need for sustainable biological control strategies.

Plant-growth-promoting rhizobacteria (PGPR) belonging to the genus *Bacillus* enhance plant growth and suppress *Fusarium oxysporum* (Fo) through multifaceted mechanisms [9,10,11,12]. Representative PGPR strains demonstrating this dual capacity include *B. subtilis* [9], *B. velezensis* [10], *B. amyloliquefaciens* [11], and *B. cereus* [12]. Their antagonistic activities include secreting antifungal lipopeptides (e.g., iturin, fengycin, surfactin) that disrupt fungal membrane integrity, siderophore-mediated iron competition, and the induction of systemic resistance in host plants. Critically, such PGPR strains demonstrate targeted efficacy against *Fusarium* pathogens in lily systems. For instance, *B. velezensis* strain E exhibited 86% inhibition of *F. oxysporum* mycelial growth in vitro through the synthesis of antifungal substances; simultaneously, rhizosphere application reduced root rot severity by 83% in potted *Lilium* plants. This biocontrol effect was synergized with growth promotion: strain E increased the bulb fresh weight through siderophore and IAA production, consequently promoting scale bulblet formation [13]. This dual synergy underscores the potential of *B. velezensis* in sustainable lily cultivation.

In this study, we isolated a novel strain, *Bacillus velezensis* R12, from the rhizosphere of asymptomatic Longya lily plants. It exhibited pronounced antagonism against Fol-L1-1 and demonstrated significant disease suppression alongside plant growth promotion in greenhouse trials. Genomic analysis revealed a comprehensive biosynthetic repertoire, including gene clusters for surfactin, fengycin, and the polyketide difficidin. This distinctive genomic potential, combined with its pronounced efficacy within the Longya lily system, distinguishes strain R12 from previously reported *B. velezensis* strains and highlights its promise as a tailored biocontrol agent.

## 2. Materials and Methods

### 2.1. Soil Sampling and Bacterial Isolation

Rhizosphere soil samples were collected from healthy Longya lily plants in Baishui Township, Yichun City, Jiangxi Province. The rhizosphere soil adhering to bulbs and roots was carefully collected using sterile brushes. Serial dilutions (10^−1^–10^−6^) of the soil samples were prepared in sterile saline (0.85% NaCl), plated on LB agar, and incubated at 30 °C for 24–48 h. Morphologically distinct colonies were purified through three rounds of streak-plate isolation.

### 2.2. Screening of Antagonistic Bacteria

Antagonistic bacteria were screened by the plate confrontation test. Fol was inoculated onto Potato Dextrose Agar (PDA) plates and cultured at 28 °C for 5–7 days. A mycelial plug (5 mm in diameter) of the pathogen was aseptically transferred to the center of a fresh PDA plate using a sterile cork borer (Biosharp, Langjieke Technology Co., Ltd., Hefei, China). After 1 day of culture at 28 °C, all the bacterial strains were spot-inoculated at equidistant positions 2.5 cm away from the central plug. Plates were re-incubated at 28 °C for 5 days, and inhibition zones were observed. The isolate exhibiting maximal inhibition was selected for further study. The isolation and identification of the pathogenic strain Fol L1-1 are described in detail in Appendix A.

### 2.3. Inhibitory Effect of Strain R12 Bacterial Culture Filtrate (BCF) on Spore Morphology and Hyphal Growth of Fol L1-1

Fol L1-1 spore suspension preparation: fungal cultures grown in PDB at 28 °C/180 rpm for 5 days were filtered through sterile gauze. Harvested spores were resuspended in sterile distilled water and adjusted to 1 × 10^6^ spores/mL using a hemocytometer.

R12 BCF preparation: after reviving from −80 °C storage on LB agar (37 °C, 24 h), a single colony was inoculated into LB broth (37 °C, 200 rpm, 12 h). Subcultured broth (100 μL) was transferred to fresh LB medium (28 °C, 180 rpm, 48 h), followed by centrifugation (10,000× *g*, 10 min) and sterile filtration (0.22 μm) to obtain cell-free filtrate.

Inhibition assay: Fol L1-1 spores were exposed to 10% (*v*/*v*) BCF (treatment) or LB broth (control) at 28 °C/180 rpm. After 5 h of co-incubation, spore morphological alterations were assessed using scanning electron microscopy (SEM; Chengdu SCI-GO Research Service Co., Ltd., Chengdu, Sichuan, China). Following 72 h incubation, hyphal morphological changes were analyzed by light microscopy and SEM.

### 2.4. Scanning Electron Microscopy of Strain R12

The R12 suspension was centrifuged (10,000× *g*, 10 min), pelleted cells were fixed with 2.5% glutaraldehyde (4 °C, 24 h), washed thrice with 0.1 M phosphate buffer (pH 7.4), and processed for imaging using a Nova Nano SEM 450 microscope (FEI, Hillsboro, OR, USA).

### 2.5. Greenhouse Biocontrol Efficacy of Strain R12 Against Longya Lily Fusarium Wilt

Pathogen preparation: Fol L1-1 spore suspensions were prepared as previously described, with concentration adjusted to 1 × 10^7^ CFU/mL using sterile distilled water.

Biocontrol agent preparation: Strain R12 was cultivated according to the BCF preparation protocol detailed previously (28 °C, 180 rpm, 24 h in LB broth). Cultures were centrifuged (10,000× *g*, 10 min), washed, and resuspended in sterile water to 1 × 10^7^ CFU/mL.

Experimental design: Longya lily bulb scales were surface-sterilized in 75% (*v*/*v*) ethanol for 30 s, followed by three rinses with sterile water. After air-drying, each bulb scale was planted individually in a pot (16.3 cm in height, 20 cm in diameter) containing 500 g of sterilized soil mixture. Plants were cultivated in a greenhouse maintained at 25 °C under a 16 h light/8 h dark photoperiod. After 10 days, the seedlings were subjected to the following treatments: (1) drenching with 15 mL of strain R12 suspension; (2) drenching with 15 mL of strain R12 suspension followed 24 h later by 15 mL of Fol L1-1 spore suspension; (3) drenching with 15 mL of 0.1% hymexazol followed 24 h later by 15 mL of Fol L1-1 spore suspension; (4) drenching with 15 mL of sterile distilled water; or (5) drenching with 15 mL of sterile distilled water followed 24 h later by 15 mL of Fol L1-1 spore suspension. Each treatment was performed using 10 plants with 3 replicates. At 45 days post-inoculation with R12, shoot fresh and dry weights were measured. For plants inoculated with Fol L1-1, disease severity and disease index were recorded 45 days after R12 inoculation. Disease severity was rated on a 0–4 disease scale: 0 = no symptoms; 1 = leaf wilting affecting less than 1/4 of the seedling; 2 = leaf wilting affecting 1/4 to 1/2 of the seedling; 3 = leaf wilting affecting more than 1/2 of the seedling; 4 = complete wilting and death of the plant. The disease index (DI) was calculated as follows: DI = [[(0 × N_0_) + (1 × N_1_) + (2 × N_2_) + (3 × N_3_) + (4 × N_4_)]/T × 4] × 100, where N_0_–N_4_ represent the number of plants in each disease rating category, and T is the total number of plants assessed. Disease incidence was calculated as: [N_1_ + N_2_ + N_3_ + N_4_]/T × 100%. Control efficacy was determined as: (DI of control − DI of treatment)/DI of control × 100%.

### 2.6. Plant Growth-Promoting (PGP) Trait Analysis of Strain R12

The PGP capabilities of strain R12 were evaluated through quantitative assays for phosphate solubilization, siderophore production, and indole-3-acetic acid (IAA) synthesis. Phosphate solubilization was assessed per Nautiyal [14] via halo formation on Pikovskaya’s agar (7 d, 30 °C), siderophore production detected on CAS agar [15] through orange halo measurement after 72 h incubation, and IAA synthesis quantified colorimetrically [16] using Salkowski reagent (530 nm) against standards after culturing R12 in LB + 0.2% L-tryptophan (48 h, 30 °C). All assays were conducted with triplicate biological replicates with three technical repetitions each.

### 2.7. Identification of Antagonistic Bacteria R12

Morphological characterization: Morpho-physiological characterization was conducted following Bergey’s Manual of Systematic Bacteriology, including biochemical tests (starch hydrolysis, gelatin liquefaction, methyl red, nitrate reduction), enzymatic assays (oxidase, catalase, Voges-Proskauer), and metabolic trait evaluation (indole production, H_2_S generation).

16S rRNA gene identification: Genomic DNA was extracted from strain R12, followed by PCR amplification of the 16S rDNA using universal primers 27F (5′-AGAGTTTATCCTGGGCTCAG-3′) and 1492R (5′-GGTTACCTTGTTACGACTT-3′). The purified PCR products were subsequently sequenced by Sangon Biotech Co., Ltd. (Shanghai, China). The obtained sequences were subjected to BLAST (version 2.16.0) homology alignment against the NCBI database. Phylogenetic tree construction was performed using the neighbor-joining method implemented in MEGA 11.0 software.

### 2.8. Genome Sequencing and Assembly of Strain R12

Whole-genome sequencing of DNA samples was performed by Baimaike Biotechnology Co., Ltd. (Beijing, China). Following quality control, high-molecular-weight DNA was fragmented using g-TUBE devices (Covaris, LLC, Woburn, MA, USA). The sheared DNA was subjected to damage repair and end—repair procedures, followed by ligation with dumbbell—shaped adapters. The resulting products were treated with exonuclease to remove unligated fragments. Finally, target fragments were size—selected using the BluePippin system (software v.6.31, Sage Science, Beverly, MA, USA) to construct the sequencing library.

The filtered reads were assembled using Hifiasm v0.12. Circularization and origin positioning were then performed using Circlator v1.5.5. The genome was further polished using Pilon v1.22 with second—generation sequencing data to achieve a higher—accuracy genome assembly. A circular genome map of R12 was generated using Circos v0.66.

### 2.9. Genomic Functional Annotation of Strain R12

The predicted sequences of the genes underwent BLAST analysis against multiple databases, including Nr, KEGG, eggNOG, SwissProt, TrEMBL, TCDB, and PHI-base, to obtain functional annotation results. Based on the BLAST results from the Nr database, the genomic sequences were annotated against the Gene Ontology (GO) database using Blast2GO software (v2.5). Sequences of unknown function were compared with the Pfam database using HMMER software (v3.4) to complete Pfam database annotation. Additionally, the protein sequences were aligned with Hidden Markov Models (HMMs) of each family in the CAZy database using HMMER software to accomplish CAZy database annotation. The Resistance Gene Identifier (RGI) tool from the Comprehensive Antibiotic Resistance Database (CARD) was utilized to align the protein sequences with the database, generating CARD database annotation results. Finally, the protein sequences were subjected to BLAST alignment against the core dataset Sets A to obtain the Virulence Factors Database (VFDB) annotations for the genome.

### 2.10. Analysis of Gene Clusters Responsible for the Biosynthesis of Secondary Metabolites

Potential secondary-metabolite biosynthetic gene clusters were systematically mined and annotated with antiSMASH 8.0.3, following the pipeline described by Blin et al. [17].

### 2.11. Statistical Analyses

All data were processed and analyzed using Microsoft Excel and IBM SPSS Statistics (version 26), respectively. Statistical significance among groups was determined by one-way analysis of variance (ANOVA) followed by the Least Significant Difference (LSD) post hoc test for multiple comparisons. Data are presented as the mean ± standard error of the mean (SEM) from three independent biological replicates (*n* = 3). Differences were considered statistically significant at *p* < 0.05. All graphs were generated using Origin 2024 software.

## 3. Results

### 3.1. Isolation and Screening of Antagonistic Bacteria Against Fusarium oxysporum

A total of 110 bacterial strains exhibiting distinct colony morphologies were isolated and purified from rhizosphere soil of asymptomatic Longya lily plants. In dual-culture assays with *Fusarium oxysporum* f. sp. *Lilii* strain L1-1 on PDA medium, four strains significantly inhibited mycelial growth [7]. Among these, strain R12 exhibited the strongest antagonistic activity against Fol L1-1 (Figure 1A,B). This strain’s biocontrol activity is not limited to a single pathogen; our previous work has demonstrated its efficacy against a spectrum of fungi, including *Fusarium* sp. L3, *Penicillium ochrochloron* F5, and *Penicillium janthinellum* F7 [7].

### 3.2. Antagonistic Activity of Strain R12 BCF Against Fol L1-1

Morphological alterations in Fol L1-1 conidia following 5 h exposure to R12 BCF are shown in Figure 2. SEM analysis revealed that untreated conidia maintained turgidity with intact surfaces, whereas R12 BCF-treated conidia exhibited severe deformation, including surface invagination and cellular shrinkage. Additionally, R12 BCF induced aberrant hyphal development: control hyphae displayed robust growth with uniform diameter and smooth morphology, while co-cultured hyphae showed ultrastructural collapse, featuring irregular invaginations, localized disintegration, and fragmentation. These results demonstrate that R12 BCF directly compromises the structural integrity of Fol L1-1 conidia and hyphae, leading to growth suppression.

### 3.3. Physiology, Biochemistry, and 16S rRNA Gene Identification of Antagonistic Bacterial Strain R12

Morphological characterization: After 12 h of incubation on LB agar plates, colonies of strain R12 exhibited rough texture, desiccated surfaces, and wrinkled morphology. Scanning electron microscopy (SEM) revealed straight, flexible rod-shaped cells, confirming their bacillus morphology (Figure 3).

Physiological and biochemical characterization: Strain R12 is a Gram-positive bacterium. It yielded negative results for the Voges-Proskauer (V-P) reaction and mannitol utilization but was positive for all other tested traits (Table 1).

Molecular identification and phylogenetic analysis: Genomic DNA was extracted from strain R12, and the 16S rRNA gene was amplified using universal primers 27F/1492R. The sequenced PCR product was subjected to BLAST analysis, revealing 99% sequence similarity to *Bacillus velezensis* in the NCBI database. A phylogenetic tree constructed with MEGA 11.0 showed that strain R12 clusters with *Bacillus velezensis* CBMB205 on a distinct branch (Figure 4).

Integrating morphological, physiological, biochemical, and molecular evidence confirmed that antagonistic strain R12 is *Bacillus velezensis*.

### 3.4. Assessment of PGP Traits in B. velezensis R12

The PGP traits of *B. velezensis* R12, including IAA synthesis, siderophore production, and phosphate solubilization, were evaluated. The results indicated that strain R12 possesses the ability to produce IAA (Figure 5A), albeit at a relatively low concentration of 6.87 mg/L. In contrast, the strain demonstrates strong siderophore production capacity, with an A/Ar value of 0.388 and a yield reaching 32.56 mg/L (Figure 5B), but does not exhibit phosphate-solubilizing activity (Figure 5C). Therefore, the growth-promoting effects of R12 on Longya lily may be attributed to siderophore and IAA production, rather than to phosphate solubilization.

### 3.5. Effects of B. velezensis R12 on Fusarium Wilt Suppression and Growth Promotion in Longya Lily

*B. velezensis* R12, which exhibited the highest antagonistic activity against Fol L1-1 in vitro, was selected for further assessment of its biocontrol efficacy in greenhouse trials. The results indicated that R12 significantly promoted the growth of Longya lily seedlings both under Fol L1-1 challenge and in pathogen-free conditions (Figure 6). Furthermore, R12 effectively alleviated disease symptoms and reduced the severity of *Fusarium* wilt at 45 days after inoculation with Fol L1-1. Data on disease incidence, disease index, and biocontrol efficacy confirmed that R12 substantially suppressed wilt symptoms caused by Fol L1-1 (Table 2).

### 3.6. Genome Sequencing and Assembly of R12

Whole-genome sequencing of strain R12 was sequenced and assembled using PacBio Sequel II platform single-molecule real-time (SMRT) technology. The genome map showed that 22,709 gene sequences were obtained. Sequencing yielded 210,960,009 bp of clean data, and the length of N50 was 10,170 bp. The final assembly revealed a complete circular chromosome of 4,015,523 bp with a GC content of 46.42% (Figure 7 and Table 3).

### 3.7. Functional Annotation of the Genome of R12

Comparative genomic annotation of strain R12 against seven functional databases revealed differential gene assignment patterns: 3803 protein-coding genes were mapped to NCBI non-redundant (Nr), 2993 to Gene Ontology (GO), 2171 to Kyoto Encyclopedia of Genes and Genomes (KEGG), 3024 to eggNOG, 3392 to Pfam, 2829 to Swiss-Prot, and 3798 to TrEMBL databases (Appendix A).

### 3.8. Annotation Results Description of the Nr Database

The Nr database (Non-Redundant Protein Database) consolidates experimentally validated and computationally predicted protein sequences with taxonomic classification. Comparative analysis indicated that the Nr database achieved the highest annotation rate among the evaluated repositories. In this study, 3803 genes in the R12 genome were functionally annotated via the Nr database, with 62.32% showing homology to characterized *Bacillus* species (Figure 8).

### 3.9. Description of GO Database Annotation Results

Gene Ontology (GO) annotation categorizes biological functions into three primary domains: cellular component, molecular function, and biological process. Functional annotation of strain R12 against the GO database assigned 2993 protein-coding genes, with hierarchical distribution as follows: 1488 genes annotated to cellular component organization, 2372 genes to molecular function (constituting 79.3% of total annotations), and 2200 genes to biological process regulation (Figure 9).

### 3.10. Annotation Results Description of the eggNOG Database

BLAST alignment of R12-encoded proteins against the eggNOG database annotated 3024 genes with functional orthologous groups. The predominant functional categories comprised: amino acid transport and metabolism (283 genes, 9.18%); general function prediction (276 genes, 8.96%); transcription regulation (235 genes, 7.62%), and carbohydrate transport and metabolism (225 genes, 7.30%). Chromatin structure and dynamics represented the least populated category (1 gene, 0.03%) (Figure 10).

### 3.11. Genomic Analysis of Antimicrobial Biosynthetic Potential in R12

Online prediction of the secondary metabolite biosynthesis gene clusters in *B. velezensis* R12 was performed using the antiSMASH 8.0.3 database, which revealed high sequence homology to known biosynthetic gene clusters. As shown in Appendix A, a total of 15 secondary metabolite gene clusters were identified in the genome of strain R12. Among these, eight clusters exhibited high genetic similarity to those encoding known bioactive compounds, including surfactin, macrolactin H, bacillaene, fengycin, difficidin, bacillibactin, bacilysin, and mersacidin.

Genomic analysis revealed that R12 possesses complete biosynthetic gene clusters for key lipopeptides, including surfactin, fengycin, and difficidin. These compounds are well-established to act synergistically in membrane disruption, surfactin facilitating pore formation that enhances fengycin’s membrane permeabilization [8], while difficidin exhibits broad-spectrum antibacterial activity. Surfactin promotes biofilm matrix assembly in *Bacillus* spp. through quorum sensing modulation [18] and demonstrates dual biocontrol efficacy via induced systemic resistance (ISR) activation in plants and direct antifungal activity [19,20,21]. Macrolactin can inhibit protein synthesis in bacteria and possesses potent antibacterial activity [22]. Bacilysin, a dipeptide antibiotic, exerts activity through its C-terminal L-antipain domain, though its biosynthesis is strain-specific in *Bacillus* [23]. Bacillomycin lyses hyphal and conidial cell walls via surfactant activity [24] while inducing lethal reactive oxygen species (ROS) accumulation in fungal propagules [25].

We hypothesize that the synergistic action of surfactin and fengycin gene clusters may be responsible for the observed membrane damage, though this requires experimental validation through gene knockout studies. Further investigations are required to delineate the dominant lipopeptide governing L1-1 antagonism and its molecular mechanism.

## 4. Discussion

Lily bulb rot disease progressively induces tissue maceration and complete bulb decomposition. Our experiments demonstrated that both vegetative cells of *B. velezensis* R12 and its cell-free fermentation supernatant exhibited pronounced antagonism against *Fusarium oxysporum* L1-1. Previous studies have established B. velezensis’s biocontrol efficacy across plant-pathogen systems. Notably, Khan et al. [26] isolated *B. velezensis* Lle-9 from lily bulbs, demonstrating dual functionality through antifungal activity and phytohormone-mediated growth promotion. Genomic analyses by Li et al. [13] further corroborated *B. velezensis*’s genomic repertoire for integrated disease management, functioning simultaneously as a rhizosphere-competent biocontrol agent and nitrogen-fixing biofertilizer in lily cultivation systems.

The broad-spectrum biocontrol potential of *Bacillus* spp. against diverse phytopathogens is well-documented. Khanal et al. [27] isolated *Bacillus* sp. ST24 and *Bacillus panaciterrae* ST25 from rice seeds, demonstrating antagonistic activity against *Marasmius graminum* and *Rhizoctonia solani* AG11, causal agents of rice seedling blight. Degani et al. [28] confirmed the inhibitory effects of *Bacillus subtilis* R2 on *Magnaporthiopsis maydis*, the etiological agent of maize late wilt disease. Wei et al. [29] found that *Bacillus mojavensis* ZA1 can produce antimicrobial substances that inhibit *Colletotrichum coccodes*. Kamalanathan et al. [30] reported nematicidal activity of *Bacillus velezensis* VB7 against *Meloidogyne incognita* (root-knot nematode) in tomato. Shuang et al. [31] isolated *Bacillus* sp. K-9 from the interface between lesions and healthy tissue on potato tubers, and demonstrated its potent suppression of *Rhizoctonia canker*. Cui et al. [32] characterized *Bacillus amyloliquefaciens* 3–5 from potato tubers, revealing its efficacy against potato common scab.

*Bacillus* spp. show significant potential for controlling infectious diseases in Liliaceae plants. Wu et al. [33] established that *Bacillus agavensis* LFM-30 effectively suppresses *Fusarium*-mediated wilt in Liliaceae hosts. Tu et al. [34] reported biocontrol efficacies of 58.74% and 68.93% for *Bacillus amyloliquefaciens* BF1 and *Bacillus subtilis* Y37 against lily wilt disease, respectively. Li et al. [13] isolated *Bacillus velezensis* strain E from lily rhizospheric soils, which exhibited potent antagonism against *Fusarium oxysporum* and *Botrytis cinerea*. Liu et al. [35] elucidated that *Bacillus* cereus produces induced systemic resistance (ISR)-activating elicitors to suppress foliar pathogens in lilies through systemic acquired resistance pathways.

*Bacillus velezensis* has been widely recognized as a biocontrol agent with broad-spectrum antimicrobial activity. Yao et al. [36] reported its ovicidal and larvicidal efficacy against *Meloidogyne incognita*; R et al. [37] demonstrated in vitro and in silico inhibition of *Fusarium oxysporum* by *B. velezensis* CBMB205; Zhao et al. [38] validated *B. velezensis* JIN4’s suppression of *Pseudomonas syringae*-induced kiwifruit bacterial canker. Additional studies confirmed its biocontrol potential against rubber tree red root rot [39], apple anthracnose [40], postharvest chili anthracnose [41], rice sheath blight disease [42], and cereal diseases [43]. However, the antifungal mechanisms of *B. velezensis* [44] and its pan-genomic characteristics [45,46,47,48] remain inadequately characterized. This is particularly true for its antagonistic interactions with lily bulb rot pathogens, which are poorly understood.

*Bacillus velezensis* R12 possesses a unique complement of biosynthetic gene clusters (BGCs), including those for difficidin and bacillaene, which underpins its experimentally verified dual functionality—suppressing *Fusarium* wilt while promoting plant growth in Longya lily. Our genomic analysis revealed the presence of complete biosynthetic gene clusters for known antifungal lipopeptides such as surfactin, fengycin, and iturin. While the observed biocontrol efficacy is consistent with the activity of these compounds, future work will include liquid chromatography-mass spectrometry (LC-MS) analysis to experimentally validate the production and quantify the levels of these metabolites by strain R12 under various conditions. Functional annotation via the eggNOG database assigned 3024 genes to secondary metabolite biosynthesis, transport, and metabolic pathways, providing a genetic foundation for these observed activities. These findings establish a basis for the subsequent functional characterization of these BGCs and the antimicrobial metabolites they encode in *B. velezensis* R12.

While the results from this study are promising, the path to practical application requires further validation. Future research will focus on large-scale field trials to confirm efficacy under natural conditions, testing against a wider array of regionally relevant pathogen strains, and direct metabolite confirmation through advanced analytical chemistry.

## 5. Conclusions

The antagonistic bacterium *Bacillus velezensis* R12 demonstrated significant biocontrol potential against lily bulb rot disease, effectively inhibiting hyphal growth and spore germination of *Fusarium oxysporum* L1-1. Genomic analysis revealed a complete circular chromosome of 4,015,523 bp harboring diverse biosynthetic gene clusters for antimicrobial secondary metabolites, which underpin its efficacy and highlight its promise for development into a sustainable biocontrol agent. Future work will focus on validating the functional roles of specific biosynthetic gene clusters through gene expression studies and targeted mutagenesis.

## Figures and Tables

**Figure 1 microorganisms-13-02430-f001:**
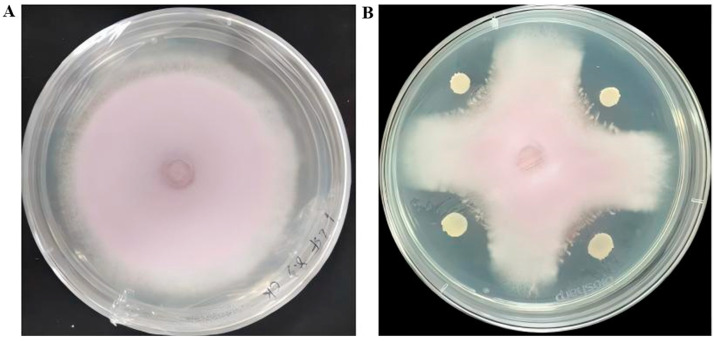
Antagonistic activity of strain R12 against *Fusarium oxysporum* f. sp. *lilii* L1-1. (**A**) Inhibition zone formed by strain R12; (**B**) Control plate without bacterial inoculation.

**Figure 2 microorganisms-13-02430-f002:**
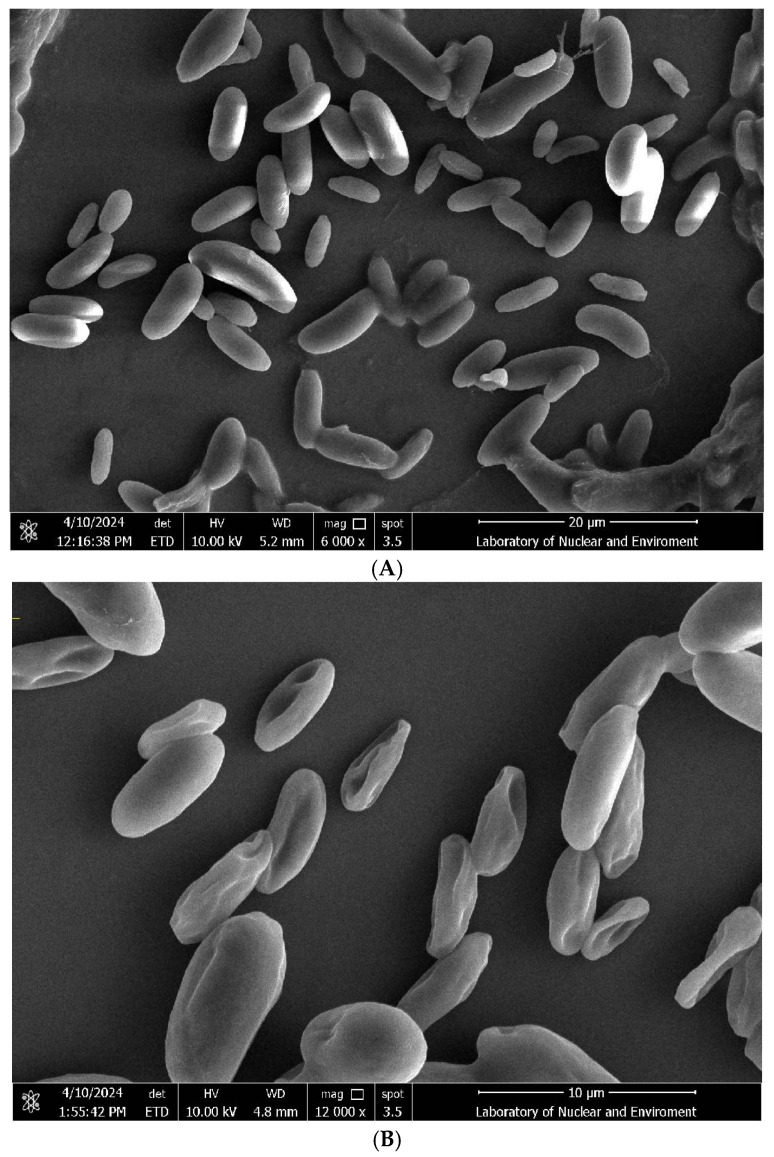
SEM analysis of morphological alterations in Fol L1-1 conidia and hyphae following exposure to R12 BCF. (**A**) Untreated Fol L1-1 conidia; (**B**) R12 BCF-treated conidia; (**C**) Untreated Fol L1-1 hyphae; (**D**) R12 BCF-treated hyphae.

**Figure 3 microorganisms-13-02430-f003:**
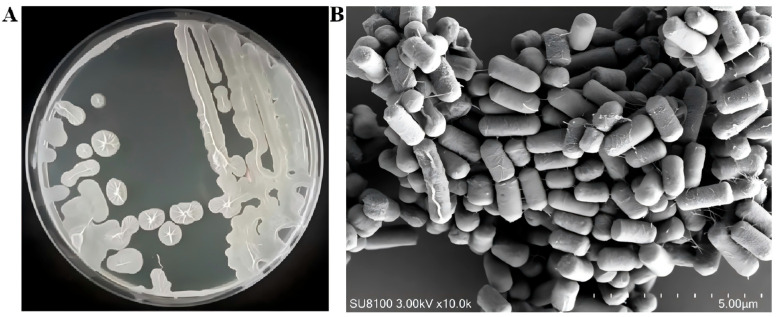
Morphology of strain R12. (**A**) Colony morphology of strain R12; (**B**) SEM analysis of strain R12 morphology.

**Figure 4 microorganisms-13-02430-f004:**
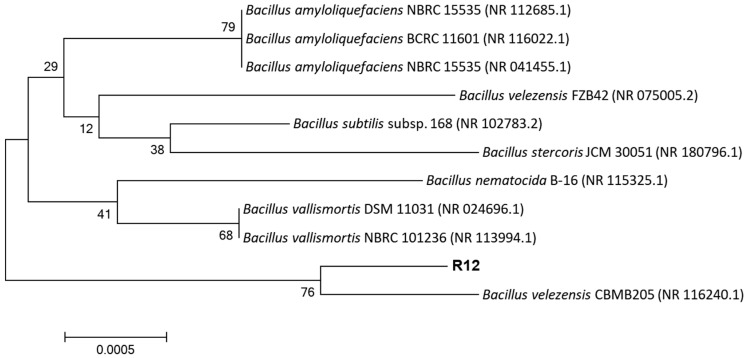
Phylogenetic Tree of Antagonistic Bacterium R12.

**Figure 5 microorganisms-13-02430-f005:**
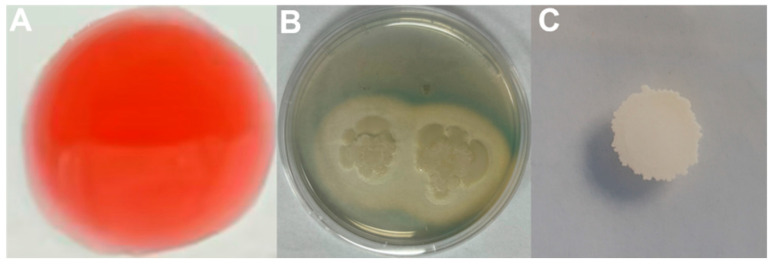
PGP traits of *B. velezensis* R12. (**A**) IAA production; (**B**) siderophore production; (**C**) phosphate solubilization (*n* = 3 biological replicates).

**Figure 6 microorganisms-13-02430-f006:**
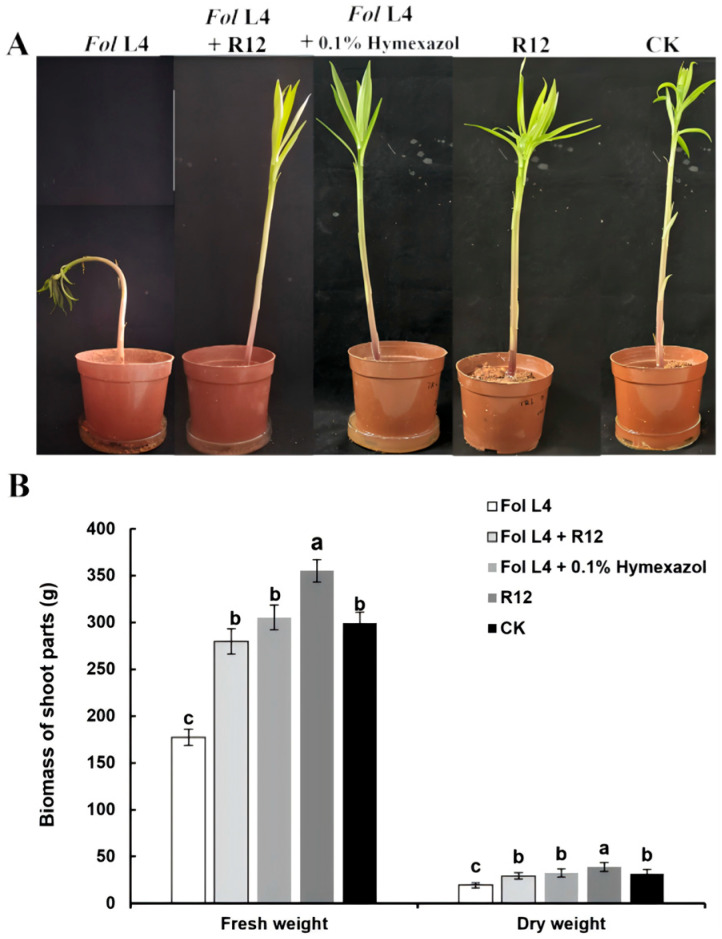
Biocontrol of *Fusarium* wilt and growth promotion of Longya lily seedlings by *B. velezensis* R12. (**A**) phenotypic representation of Longya lily seedlings across different treatments (*n* = 3 biological replicates); (**B**) Biomass of Longya lily seedlings under different treatments. Data represent mean ± SD (*n* = 3), different letters indicate significant differences at *p* < 0.05 level.

**Figure 7 microorganisms-13-02430-f007:**
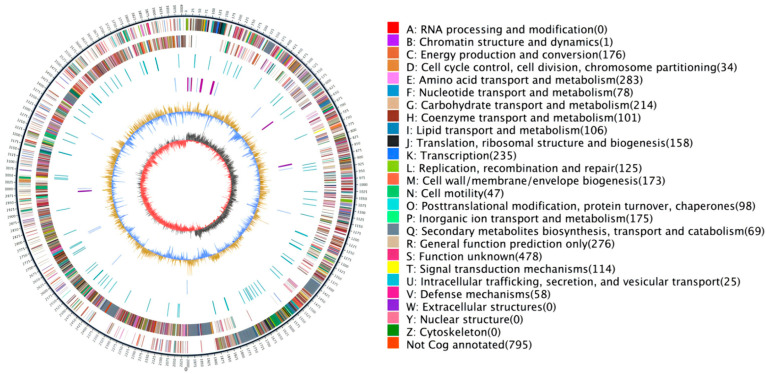
Circular genome map of *B. velezensis* R12. The outermost circle indicates genome size, with each mark representing 5 kb. The 2nd and 3rd circles show genes on the positive and negative strands, respectively, where different colors indicate different COG functional categories. The 4th circle shows repeated sequences. The 5th circle shows tRNA (blue) and rRNA (purple). The 6th circle shows GC content; light yellow areas indicate GC content above the genome average, with higher peaks showing greater differences, while blue areas indicate GC content below the average. The innermost circle shows GC-skew, with dark gray showing regions where G content is higher than C, and red showing regions where C content is higher than G.

**Figure 8 microorganisms-13-02430-f008:**
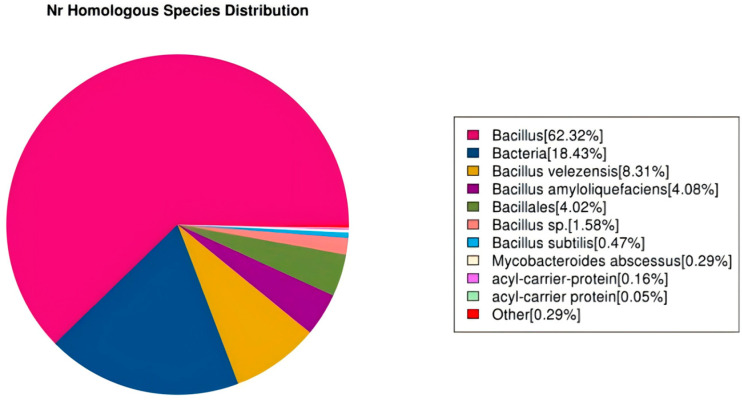
Species distribution of sequences aligned to the Nr database. This figure reflects the species distribution of sequences aligned in the Nr database. Different colors indicate distinct species.

**Figure 9 microorganisms-13-02430-f009:**
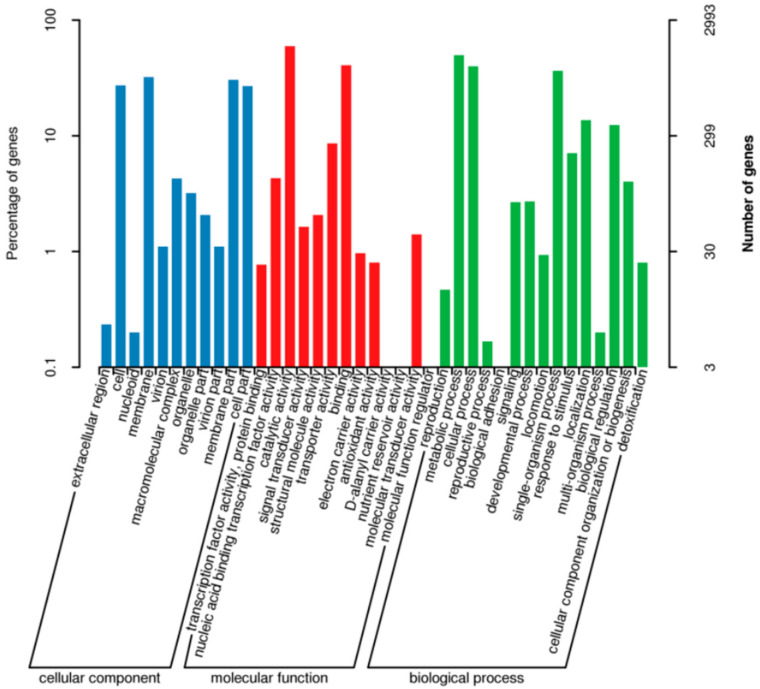
GO functional annotation classification statistics. The x-axis indicates GO subcategories, while the left y-axis represents the percentage of annotated genes and the right y-axis denotes absolute gene counts. This visualization demonstrates the enrichment patterns of level 2 GO functional subcategories relative to the whole genomic background, highlighting the dominance hierarchy among these functional subdivisions.

**Figure 10 microorganisms-13-02430-f010:**
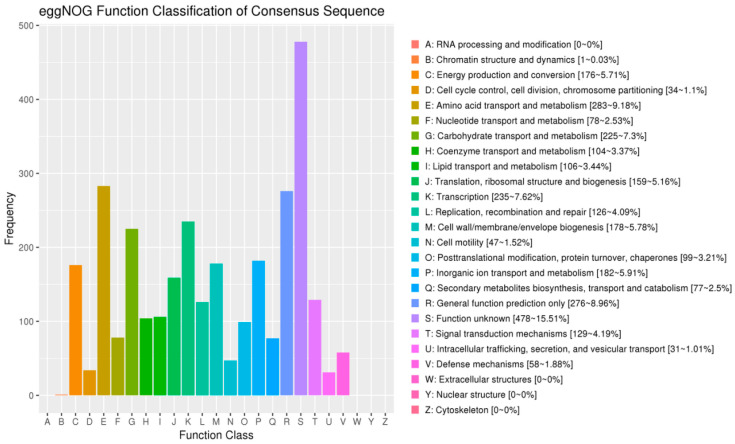
eggNOG functional gene categorization statistics. The x-axis denotes eggNOG subcategories, and the y-axis represents the relative abundance (%) of annotated genes within corresponding functional classifications.

**Table 1 microorganisms-13-02430-t001:** Physiological and biochemical characteristics of antagonist R12.

Characteristics	Results
Gram stain	+
Starch hydrolysis experiment	+
Glucose	+
Arabinose	+
Gelatine	+
Hydrogen peroxide enzyme test	+
Indole reaction	+
V-P reaction	−
mannitol	−

Note: “+” indicates positive, “−” indicates negative.

**Table 2 microorganisms-13-02430-t002:** Disease incidence, disease index, and control efficacy of *B. velezensis* R12 against Longya lily *Fusarium* wilt.

Treatment	Disease Incidence (%)	Disease Index	Control Efficacy (%)
Fol L1-1	97.26 ± 0.67 a	94.57 ± 0.51 a	
Fol L1-1 + 0.1% Hymexazol	2.19 ± 0.14 b	2.67 ± 0.13 b	97.18 ± 0.13 a
Fol L1-1 + R12	2.68 ± 0.15 b	3.77 ± 0.11 b	96.01 ± 0.07 a

Note: The data are presented as mean ± standard deviation (SD). Within the same column, different letters (a,b) indicate significant differences at *p* < 0.05 level.

**Table 3 microorganisms-13-02430-t003:** Cleaned genomic sequence data of *Bacillus velezensis* R12.

Genome		Genome Assembly	
Number of sequences	22,709	Scaffold length (bp)	4,015,523
SumBase	210,960,009	Scaffold number	1
N50 reads length (bp)	10,170	Scaffold N50 (bp)	4,015,523
N90 reads length (bp)	6141	Scaffold N90 (bp)	4,015,523
Mean reads length (bp)	9289	Contig Length (bp)	4,015,523
Max length (bp)	30,021	GC content (%)	46.42
Mean reads quality	33.73	Gaps number	0

## Data Availability

The original data presented in the study are openly available in China Center for Type Culture Collection (CCTCC) at CCTCC M 20241785.

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
