# Peer review of "Screening and Genomic Analysis of *Bacillus velezensis* R12 as a Biocontrol Agent Against *Fusarium oxysporum* Causing Wilt in Longya Lily (*Lilium brownii* var. *viridulum*)"

_microorganisms, 2025, doi:10.3390/microorganisms13112430_

Round 1
Reviewer 1 Report
Comments and Suggestions for Authors
This is a scientifically sound and potentially impactful study that contributes to sustainable disease management in Lilium brownii. However, the manuscript requires major revisions in terms of clarity, language, data consistency, and strengthening the comparative discussion before it can be considered for publication.

Author Response
Thanks for your comments. We have modified our manuscript according to your suggestions, the details are as followed:
Comments 1: The abstract is informative but somewhat lengthy. It should be more concise, focusing on key findings (antagonistic activity, growth promotion, genomic potential).
Response 1: We have thoroughly revised the abstract to make it more concise while highlighting the key findings as recommended. The modifications include:Removing redundant descriptions of pathological effects; Streamlining the methodological descriptions; Focusing on the three key aspects: antagonistic activity, growth promotion, and genomic potential and strengthening the concluding statement to better reflect the significance of our findings, as detailed in the abstract.As shown in lines 24-32.
Comments 2: Minor language polishing is needed for smoother readability.
Response 2: We agree with this comment and we have polished the language of the manuscript to enhance its readability.(eg: change "Plant-growth-promoting rhizobacteria (PGPR) of the genus Bacillus simultaneously enhance plant growth and suppress Fusarium oxysporum (Fo) through multifaceted mechanisms" to "Plant-growth-promoting rhizobacteria (PGPR) belonging to the genus Bacillus enhance plant growth and suppress Fusarium oxysporum (Fo) through multifaceted mechanisms"; and change “The DNA samples were submitted to Baimaike Biotechnology Co., Ltd. (Beijing, China) for whole - genome sequencing. After passing the quality inspection, the DNA samples were immediately sheared using a g - TUBE.” to “Whole-genome sequencing of DNA samples was performed by Baimaike Biotechnology Co., Ltd. (Beijing, China). Following quality control, high-molecular-weight DNA was fragmented using g-TUBE devices.”)
Comments 3: In the introduction section, the background on Lilium brownii and its economic/medicinal importance is well presented.
Response 3: Thank you for your positive feedback.
Comments 4: The problem of Fusarium wilt is clearly stated, but the introduction could be shortened by reducing repetition on pathogen aggressiveness and fungicide limitations.
Response 4: We agree with this comment. The relevant repetitive content has been streamlined as detailed in lines 48-55.
Comments 5: A clearer statement of novelty (why strain R12 is unique compared to previously reported B. velezensis strains) should be highlighted at the end of the introduction.
Response 5: Thank you for pointing this out. R12 exhibited pronounced antagonism against Fol-L1-1 and demonstrated significant disease suppression alongside plant growth promotion in greenhouse trials. Genomic analysis revealed a comprehensive biosynthetic repertoire, including gene clusters for surfactin, fengycin, and the polyketide difficidin. This distinctive genomic potential, combined with its pronounced efficacy within the Longya lily system, distinguishes strain R12 from previously reported B. velezensis strains and highlights its promise as a tailored biocontrol agent. We have explicitly highlighted this section in lines 71-78.
Comments 6: In the material and method section, some descriptions are too lengthy (e.g., soil sampling, SEM preparation). Methods could be streamlined for clarity.
Response 6: We agree with this comment. We have rewritten the conclusions to make this part more concise, as shown in lines 82-86 and 115-117.
Comments 7: Replication details should be consistently reported (e.g., number of biological replicates for IAA, siderophore, and phosphate solubilization assays) in the method section.
Response 7: Thank you for pointing this out. We have added the number of biological replicates: All assays conducted with triplicate biological replicates with three technical repetitions each. As shown in lines 157-158.
Comments 8:The statistical section mentions ANOVA and LSD, but exact parameters (n, error type) are not always clearly stated.
Response 8: Thank you for pointing this out. We have We have added the exact parameters in section 2.11 “Statistical Analyses”, as detailed in lines 209-215.
Comments 9:The results are comprehensive and well-structured, but there is redundancy (e.g., colony morphology and SEM details appear in both Results 3.3 and Discussion).
Response 9: Agree. We have removed the redundant descriptions of colonial morphology and SEM details that were duplicated between the Discussion and Results sections, as detailed in lines 246-258 and 458-470.
Comments 10:Figures should be checked for clarity and resolution. Legends need more descriptive detail (sample size, statistical test, significance notation).
Response 10: Thank you for pointing this out. We have improved the clarity and resolution of Figures 1 through 10, and the figure legends have been supplemented with more detailed descriptions, as shown in lines 296-297 and 314-317.
Comments 11:Genome size data are inconsistent (reported as 210,960,009 bp and 4,015,523 bp in different sections). This discrepancy must be resolved.
Response 11: Thank you for pointing this out. The genome size has been corrected to 4,015,523 bp in the manuscript, as shown in line 480.
Comments 12:Some tables (e.g., annotation statistics) could be moved to supplementary materials to improve flow.
Response 12: Agree. We have moved "Table 4. General database annotation statistics" and "Table 5. Prediction of secondary metabolite biosynthetic gene clusters in R12" to the supplementary materials.
Comments 13: The discussion is extensive but contains repetition of results. It should focus more on interpreting findings considering previous studies.
Response 13: We agree with this comment. We have streamlined the Discussion section by removing content that duplicated the Results, so that it now focuses on interpreting the findings in the context of previous studies, as detailed in lines 458-470.
Comments 14: While comparisons with other Bacillus strains are made, the unique contributions of strain R12 (e.g., secondary metabolite clusters, dual activity in lily systems) need stronger emphasis.
Response 14: We agree with this comment. We have emphasized the uniqueness of strain R12 and its dual activity in the lily system in lines 458-461.
Comments 15: The role of specific biosynthetic gene clusters (surfactin, fengycin, difficidin, etc.) in antagonism could be more critically analyzed.
Response 15: We agree with this comment. We have added the role of specific biosynthetic gene clusters in antagonism as shown in lines 398-402.
Comments 16: Some references are relevant, but others appear tangential. A more focused citation strategy would strengthen the discussion.
Response 16: Thank you for pointing this out. We have revised some of the references to ensure their relevance to the main text, including the removal of original references [29], [36], and [42].
Comments 17: Conclusions are repetitive and too long. They should be concise, summarizing the main findings and future application potential.
Response 17: Thank you for pointing this out. We have rewritten the Conclusions section to make it more concise while summarizing the main findings and future application potential, as detailed in lines 477-484.
Comments 18: Any speculative claims should be toned down or clearly labeled as hypotheses for future validation.
Response 18: Thank you for pointing this out. We have explicitly labeled speculative statements as hypotheses awaiting validation, including the addition of "We hypothesize that the synergistic action of surfactin and fengycin gene clusters may be responsible for the observed membrane damage, though this requires experimental validation through gene knockout studies." in lines 411-413, and "Future work will focus on validating the functional roles of specific biosynthetic gene clusters through gene expression studies and targeted mutagenesis." in lines 482-484.
Comments 19:The manuscript contains grammatical errors, awkward sentence structures, and redundancy. A thorough English editing is required before acceptance.
Response 19: Thank you for pointing this out. We have corrected grammatical errors in the manuscript, removed redundant content, and conducted comprehensive language polishing throughout the article.(eg: changed “Extensive studies have validated the biocontrol potential of Bacillus spp. against phytopathogens.” to “The broad-spectrum biocontrol potential of Bacillus spp. against diverse phytopathogens is well-documented.” and changed “Shuang et al. [32] isolated Bacillus sp. K-9 from potato tuber lesion-healthy tissue interfaces, showing potent suppression of Rhizoctonia canker.” to “Shuang et al. [32] isolated Bacillus sp. K-9 from the interface between lesions and healthy tissue on potato tubers, and demonstrated its potent suppression of Rhizoctonia canker.”
Comments 20: Long sentences should be simplified for clarity
Response 20: Thank you for pointing this out. We have simplified the overly long sentences to ensure clarity of expression.(eg: changed “ Sequencing results were subjected to BLAST alignment ...revealed clustering of strain R12 and Bacillus velezensis CBMB205 on the same branch” to “ The sequenced PCR product was subjected to BLAST analysis...with Bacillus velezensis CBMB205 on a distinct branch ” and changed “Our experimental results demonstrated that both B. velezensis R12 vegetative cells and their cell-free fermented supernatant exhibited pronounced antagonistic effects against Fusarium oxysporum L1-1.” to “Our experiments demonstrated that both vegetative cells of B. velezensis R12 and its cell-free fermentation supernatant exhibited pronounced antagonism against Fusarium oxysporum L1-1”)
Comments 21: This is a scientifically sound and potentially impactful study that contributes to sustainable disease management in Lilium brownii. However, the manuscript requires major revisions in terms of clarity, language, data consistency, and strengthening the comparative discussion before it can be considered for publication.
Response 21: We sincerely thank the reviewer for the positive assessment of our study's significance and for the constructive feedback. We have thoroughly addressed all the concerns raised through major revisions to the manuscript. The key improvements include the entire manuscript has been polished to enhance linguistic expression and logical clarity, with standardized data presentation and a strengthened comparative discussion.Thank you for the opportunity to improve our study.
Other minor corrections
Comments i. “sereening of antagonistie baeteria” should be “screening of antagonistic bacteria”
Response i.: Thank you for pointing this out. We have corrected "sereening of antagonistie baeteria" to "screening of antagonistic bacteria".
Comments ii.: “clean dat” should be “clean data”
Response ii.: Thank you for pointing this out. We have corrected "clean dat" to "clean data".
Comments iii. : “henotypic representation” should be “phenotypic representation”
Response iii. : Thank you for pointing this out.We have corrected "henotypic representation" to "phenotypic representation".
Comments iv.: “B.velezensis” missing space after period, should be “B. velezensis”
Response iv.: Thank you for pointing this out. We have corrected "B.velezensis" to "B. velezensis".
Comments v.: “Ge-nome” should be “Genome”
Response v.: Thank you for pointing this out. We have corrected "Ge-nome" to "Genome".
Comments vi.: B. verezensis (appears once in Results 3.5 heading) should be B. velezensis.
Response vi.: Thank you for pointing this out. We have corrected "B. verezensis" to "B. velezensis".
Comments vii.: The genome size is inconsistently reported as 210,960,009 bp in Discussion and Conclusions, versus 4,015,523 bp in Results (Table 3). One is a total sequencing data yield, the other is the genome size. This should be clarified to avoid appearing as an error.
Response vii.: Thank you for pointing this out. We have corrected the genome size to 4,015,523 bp in the manuscript.
Comments viii. Terms like “Fol L1” and “Fol L3” appear; ensure consistent naming of the pathogen strain throughout.
Response viii.: Thank you for pointing this out. We have corrected “Fol L1” and “Fol L3” to "Fol L1-1" throughout the text.

Reviewer 2 Report
Comments and Suggestions for Authors
This is an interesting and relevant study that explores the potential of a rhizobacterium, Bacillus velezensis R12, as a biocontrol agent against Fusarium wilt in Longya lily.
However, several aspects require clarification and improvement before the manuscript can be considered for publication:
The manuscript does not clearly specify which molecular markers were used for pathogen identification (ITS, TEF1, or others). More details on pathogen isolation, pathogenicity confirmation, and taxonomic validation should be included.
Testing R12 against additional Fusarium strains or other pathogens would strengthen the claim of broad-spectrum biocontrol activity.
IAA production is only presented qualitatively. Quantitative measurement (e.g., µg/mL using a standard curve with Salkowski reagent) is needed to properly evaluate the growth-promoting potential of R12.
No in vitro data are presented on the activity of hymexazol against the Fusarium isolate. Such data are essential to provide a fair comparison between chemical and biological control under standardized conditions (e.g., growth inhibition on PDA medium).
Although genome analysis identified biosynthetic clusters for known compounds (surfactin, fengycin, etc.), there is no experimental validation of their expression (e.g., LC-MS metabolite profiling or transcriptomic analysis).
The conclusions are currently too strong given the scale of the experiments. Strain R12 is promising, but further validation through field trials, testing against multiple pathogen strains, and metabolite confirmation is necessary before practical application can be considered.
Author Response
Thanks for your comments. We have modified our manuscript according to your suggestions, the details are as followed:
Comments 1: The manuscript does not clearly specify which molecular markers were used for pathogen identification (ITS, TEF1, or others). More details on pathogen isolation, pathogenicity confirmation, and taxonomic validation should be included.
Response 1: We thank the reviewer for this valuable comment. In response, we have now provided a detailed description of the pathogen identification and characterization procedures in the Supplementary Materials, includes: “Isolation, Purification, and Pathogenicity Assay of Pathogenic Fungi” and “Morphological and Molecular Identification of Pathogens”.
Comments 2: Testing R12 against additional Fusarium strains or other pathogens would strengthen the claim of broad-spectrum biocontrol activity.
Response 2: We thank the reviewer for this insightful comment. We agree that demonstrating efficacy against a broader range of pathogens is crucial for claiming broad-spectrum activity. In our previous study ([7]Liang, L.L., S.; Zhu, D.;Huang, Y.;Zheng, Q.;Ji, H.;Zhan, M.;Chen, M. . The identification of pathogens of bulb rot of Lilium browni var. viridulum and screening of antagonistic bacteria. Journal of Jiangxi Normal University(Natural Science Edition) 2024, 48, 491-498, doi:10.16357/j.cnki.issn1000-5862.2024.05.08.), we have already evaluated the antagonistic potential of strain R12 against several other phytopathogens. Specifically, R12 exhibited significant inhibitory effects against Fusarium sp. L3, Penicillium ochrochloron F5, and Penicillium janthinellum F7. These results, which are now cited in the revised manuscript (lines 222-225), substantiate our claim regarding the broad-spectrum biocontrol capability of B. velezensis R12.
Comments 3: IAA production is only presented qualitatively. Quantitative measurement (e.g., µg/mL using a standard curve with Salkowski reagent) is needed to properly evaluate the growth-promoting potential of R12.
Response 3: We thank the reviewer for this important observation. We fully agree that quantitative IAA measurement would provide more precise evaluation of the strain's growth-promoting potential we have incorporated this set of data into the main text, as detailed in lines 289-293.
Comments 4: No in vitro data are presented on the activity of hymexazol against the Fusarium isolate. Such data are essential to provide a fair comparison between chemical and biological control under standardized conditions (e.g., growth inhibition on PDA medium).
Response 4: Thank you for this valuable comment. We fully agree that providing a direct comparison between the chemical and biological control agents under standardized conditions (e.g., on PDA medium) is essential for a fair and objective assessment.As you rightly pointed out, hymexazol is a widely used fungicide for soil treatment and is recognized for its broad-spectrum efficacy against Fusarium spp. This has been well-documented in multiple studies, for instance: Yang, F.; Jiang, H.; Ma, K.; Wang, X.; Liang, S.; Cai, Y.; Jing, Y.; Tian, B.; Shi, X. Genome sequencing and analysis of Bacillus velezensis VJH504 reveal biocontrol mechanism against cucumber Fusarium wilt. Frontiers in microbiology 2023, 14, 1279695, doi:10.3389/fmicb.2023.1279695.and Xu, W.; Yang, Q.; Yang, F.; Xie, X.; Goodwin, P.H.; Deng, X.; Tian, B.; Yang, L. Evaluation and Genome Analysis of Bacillus subtilis YB-04 as a Potential Biocontrol Agent Against Fusarium Wilt and Growth Promotion Agent of Cucumber. Frontiers in microbiology 2022, 13, 885430, doi:10.3389/fmicb.2022.885430.
Importantly, we have previously conducted in vitro assays to evaluate the inhibitory effect of hymexazol on the mycelial growth of our specific pathogen, Fusarium oxysporum L1 and the results consistently showed strong inhibition, confirming its potency against this isolate. We have included the result of this specific assay below for your direct evaluation.
Comments 5: Although genome analysis identified biosynthetic clusters for known compounds (surfactin, fengycin, etc.), there is no experimental validation of their expression (e.g., LC-MS metabolite profiling or transcriptomic analysis).
Response 5: We thank the reviewer for this insightful observation regarding the need for experimental validation of metabolite expression. We fully agree that LC-MS metabolite profiling and transcriptomic analysis would provide valuable functional confirmation of the biosynthetic clusters identified in our genomic analysis.We would like to clarify that the primary focus of the current manuscript is to establish the genomic potential and the phenotypic biocontrol efficacy of B. velezensis R12 against lily bulb rot. At this stage, we have genetically identified the potential for producing these compounds and correlated it with the observed strong antifungal activity.
Importantly, the experimental work to validate the expression and production of these metabolites (e.g., surfactin, fengycin) via LC-MS profiling is currently underway in our laboratory. We consider this a crucial next step and have now explicitly stated this as an important future direction in the revised manuscript, as detailed in lines 461-466.
Comments 6: The conclusions are currently too strong given the scale of the experiments. Strain R12 is promising, but further validation through field trials, testing against multiple pathogen strains, and metabolite confirmation is necessary before practical application can be considered.
Response 6: We thank the reviewer for raising this important point regarding the strength of our conclusions and the need for further validation. We agree that robust evidence from field trials and testing against multiple pathogens is crucial for assessing practical potential.
As previously detailed in our response to Comment #2, we have now demonstrated that strain R12 exhibits a broad-spectrum antifungal activity against additional pathogens beyond Fusarium oxysporum L1, including Fusarium sp. L3, Penicillium ochrochloron F5, and Penicillium janthinellum F7. We have conducted preliminary field trials to evaluate the efficacy of R12 under natural conditions. The initial results are encouraging and indicate that the application of R12 contributes to the suppression of lily bulb rot in the field. We provide below a photographic record from these trials as preliminary evidence. The comprehensive quantitative data from these field experiments are currently being processed and statistically analyzed.
In light of the ongoing data analysis and the reviewer's valid comment, we have explicitly stated in the revised Discussion that field trials and thorough metabolite profiling (e.g., LC-MS) are essential and defined as our immediate future work to translate these promising laboratory results into practical applications.as detailed in lines 471-475.

Reviewer 3 Report
Comments and Suggestions for Authors
In my opinion, manuscript ID microorganisms-3890992 submitted for review is interesting both scientifically and with regard to its application. I appreciate it when science can bring practical applications to our lives—in this case, agriculture.
The abstract and introduction are well written and I have no comments on these parts of the work.
The "Materials and Methods" section requires minor corrections. Specifically, please include the accuracy of all devices used, e.g., incubation temperature, etc. Please add the manufacturers of the reagents/culture media. Please include temperatures with a space.
The "Results" section is described thoroughly and clearly, but it also requires minor corrections. I don't see the scale bars in Figs. 2 and 3 – please correct them. Please explain in the description of Table 1 what the "+" means and in Table 5 what the "-" means. I can guess what these symbols mean, but everything should be described in detail.
Discussion and Conclusions - I have no comments.
There is a "Patents" section in the manuscript, but no exact information is provided: title, number, etc.
Author Response
Thanks for your comments. We have modified our manuscript according to your suggestions, the details are as followed:
Comments 1: The "Materials and Methods" section requires minor corrections. Specifically, please include the accuracy of all devices used, e.g., incubation temperature, etc. Please add the manufacturers of the reagents/culture media. Please include temperatures with a space.
Response 1: We thank the reviewer for these valuable technical suggestions. We have carefully revised the Materials and Methods section to address all points raised, as detailed in lines 79-155.
Comments 2: The "Results" section is described thoroughly and clearly, but it also requires minor corrections. I don't see the scale bars in Figs. 2 and 3 – please correct them. Please explain in the description of Table 1 what the "+" means and in Table 5 what the "-" means. I can guess what these symbols mean, but everything should be described in detail.
Response 2: Thank you for pointing this out. We have supplemented the meanings of "+" and "-" in Tables 1 and 5, and revised Figures 2 and 3 to reveal complete scale bars, as shown as in line 271 and Table 5 has now been moved to the Supplementary Materials as Table S2.
Comments 3: There is a "Patents" section in the manuscript, but no exact information is provided: title, number, etc.
Response 3: Thank you for pointing this out. We have removed the "Patents" section from the manuscript as suggested.

Round 2
Reviewer 2 Report
Comments and Suggestions for Authors
The authors have satisfactorily addressed most reviewer comments and improved the manuscript accordingly.